# *C. elegans* as a Powerful Tool for Cancer Screening

**DOI:** 10.3390/biomedicines10102371

**Published:** 2022-09-23

**Authors:** Eric di Luccio, Masayo Morishita, Takaaki Hirotsu

**Affiliations:** Hirotsu Bioscience Inc., 22F The New Otani Garden Court, 4-1 Kioicho Chiyoda-ku, Tokyo 102-0094, Japan

**Keywords:** *C. elegans*, N-NOSE, multi-cancer screening, primary screening, olfaction, urine

## Abstract

Regular cancer screening is critical for early cancer detection. Cancer screening tends to be burdensome, invasive, and expensive, especially for a comprehensive multi-organ check. Improving the rate and effectiveness of routine cancer screenings remain a challenge in health care. Multi-cancer early detection (MCED) is an exciting concept and a potentially effective solution for addressing current issues with routine cancer screening. In recent years, several technologies have matured for MCED, such as identifying cell-free tumor DNA in blood or using organisms such as *Caenorhabditis elegans* as a tool for early cancer detection. In Japan, N-NOSE is a commercially available multi-cancer detection test based on the chemotaxis of *C. elegans* using a urine sample showing 87.5% sensitivity and 90.2% specificity. In this review, we focus on using *C. elegans* as a powerful biosensor for universal cancer screening. We review N-NOSE clinical research results, spotlighting it as an effective primary cancer screening test.

## 1. Healthcare Utilization for Cancer Screening

### 1.1. Cancer Statistics

In 2020, in Japan, the cancer incidence was slightly over 1 million, 58% males and 42% women [1]. For males, the most common cancers were: prostate (16% of all cancers), stomach (16%), colon/rectum (15%), lung (15%), followed by liver (5%), pancreas (4%), and esophagus (4%) [1]. For women, the most common cancers were breast (21% of all cancers), colon/rectum (16%), lung (10%), stomach (10%), uterus (7%), pancreas (5%), and liver (3%). In 2020, there were 380,000 cancer deaths, 58% males and 42% women [1].

Cancer death can be prevented by early diagnosis and early access to an effective cure. Among countries, differences exist regarding health care utilization, especially regarding cancer screening. Cancer screening guidelines vary from country to country and differ within countries. Looking at the 21 countries with the highest per capita spending on healthcare reveals some commonalities in cancer screening recommendations [2]. In most countries, breast, cervical, colorectal, prostate, skin, and lung cancer are recommended for cancer screening [2]. Screening recommendations for breast cancer, cervical cancer, and to some extent, colorectal cancer is the most homogeneous among countries. However, guidelines largely differ from country to country by screening method, starting, and ending ages, along with screening interval [2]. Cancer screening programs have a significant effect on reducing cancer-specific mortality [3,4]. For example, breast cancer mortality in the United States has decreased by 53.2% between 1975 and 2014, with similar trends observed across Europe, attributable to both screening and improved treatment [5]. According to OECD statistics, in Japan in 2019, only 44.6% of females aged 50–69 were screened for breast cancer, while 83.9% were screened in Israel [6]. For colorectal cancer, in the United States in 2019, 68.9% of males ages 50–74 were screened, while only 47.8% in Japan. Improving cancer screening rates and healthcare utilization remains challenging in many countries, such as Japan. However, to assess the true importance of cancer screening programs and access to treatment, as of 2020, cancer accounts for nearly 10 million deaths per year, according to WHO statistics, with most cases occurring in low to middle-income countries [7,8]. According to WHO, the most common causes of cancer death in 2020 were: lung (1.80 million deaths), colon and rectum (916,000 deaths), liver (830,000 deaths), stomach (769,000 deaths), and breast (685,000 deaths). Each year, approximately 400,000 children develop cancer [7].

### 1.2. Cancer Screening Economics

The COVID-19 pandemic has immensely stressed the healthcare system and has drastically impacted cancer screening rates. In the United States, an estimated 9.4 million cancer screenings were skipped in 2020 alone [9]. It may inevitably lead to cancer diagnosis at a more advanced stage and increased cancer mortality.

Improving cancer screening is a multi-factorial, medical, economic, and societal problem. Most of the population usually do not spontaneously seek a cancer screening without any indication of a disorder. Due to a lack of obvious clinical signs at early stages, many cancers are unfortunately diagnosticated late. In addition, cancer screening can be physically invasive and burdensome, such as transvaginal ultrasound, colonoscopy, and gastroscopy, and may require anesthesia, which deters some individuals from undergoing these examinations. For instance, about 35% of colonoscopies are conducted in the United States with anesthesia services [10]. In addition, the economic impact of cancer is significant. In 2020, the economic burden of cancer on the U.S. healthcare system reached USD $157.7 billion [11]. Cancer also significantly impacts the financial health of patients and families, with 16.1 million cancer patients in the U.S. having out-of-pocket medical expenses and 1 in 4 having difficulties paying bills [11]. 

Screening for early cancer contributes to mitigating the economic, medical, and psychological issues related to cancer, but the cost, without insurance, remains prohibitive. For example, a simple diagnostic mammogram in the United States costs up to USD $1000. In Japan, a comprehensive full-body cancer screening by PET-CT costs JPY 132,000 (~USD 1000) without medical insurance. Despite cost variations across countries, in all cases, the cost of cancer screening, primarily a comprehensive multi-organ screening, remains prohibitively expensive for most.

## 2. Blood Tumor Markers and Liquid Biopsy

Protein tumor markers in the blood, such as carcinoembryonic antigen (CEA), cancer antigen 15-3 (CA15-3), and cancer antigen 19-9 (CA19-9), among others, are routinely used as prognosis biomarkers or cancer surveillance and monitoring response [12]. However, CEA, CA15-3, and CA19-9 have low sensitivity from aggregated results on multiple cancers and all stages considered, at 33.3%, 10.0%, and 11.8% sensitivity, respectively [13]. Cancer screening aims to detect cancer at early stages before symptoms appear to offer a chance to start treatment early, but relying on protein tumor markers may not be clinically valuable [14]. Cancer tests must show sensitivity and specificity, especially from early stage 0-I. Non-invasive tests, such as liquid biopsy, hold promise for accurately screening people for cancer and could help advance early detection. In addition to cancer screening, liquid biopsies open a new avenue for minimally invasive tumor biopsy for diagnosis. An invasive tumor biopsy, intrinsic difficulties in assessing certain tumors, and intra-tumor heterogeneity render the application of liquid biopsy, a minimally invasive approach by essence, an exciting R&D endeavor to bring to the clinic [15]. Transitioning liquid biopsy from R&D to the clinic to improve survival rates of cancer patients has attracted much attention in recent years due to the great potential of blood-based biomarkers for early diagnosis. Several circulating biomarkers, including circulating microRNAs (miRNAs), cell-free DNA (cfDNA), and circulating tumor DNA (ctDNA), for the development of tests for early cancer detection have been the basis of the development of new cancer screening tests [16,17]. Still, most cancer types lack well-established biomarkers to identify the disease, which remains a bottleneck in developing liquid biopsies [15]. The low and variable concentration of biomarkers in early-stage cancer is a challenge for effective diagnosis as different blood samples from the same individual might yield different results [15]. Low levels of biomarkers imply that liquid biopsy techniques must be highly sensitive. Detecting cfDNA/ctDNA or other biomarkers with high sensitivity can negatively affect the specificity of the test [15]. Despite significant progress, the major challenge of cfDNA/ctDNA detection remains assay sensitivity and specificity, especially for early-stage detection. In addition, false-positive could be triggered by benign mutations [18].

## 3. Multi-Cancer Early Detection (MCED)

### 3.1. A New Paradigm in Cancer Screening

Despite challenges associated with liquid biopsies, the development of multi-cancer early detection tests is the focus of much attention. Universal cancer screening, or MCED, is raising hope of having a single, minimally invasive test for simultaneous detection of multiple cancers, possibly at early stages. It is a radical departure from the classical methods, “one test, one cancer”. It is a new paradigm, and MCED could potentially help improve public health, early detect more types of cancer at the earliest stage, lower the economic and physiological burden on people from cumbersome and invasive tests and reduce healthcare expenditures. MCED is also raising hope for precision medicine to assist in accurate cancer diagnosis. Strikingly, a simulation of the long-term performance of having MCED in the healthcare arsenal could result in a drastic reduction of 26% of all cancer-related deaths in 50–79 years old in the United States and could also reduce late-stage (III, IV) incidence by 78% [19].

### 3.2. Galleri by GRAIL, a Blood-Based MCED Test with Cell-Free DNA (cfDNA)

Among new technologies developed for MCED tests, GRAIL is marketing Galleri. Using a blood sample, this multi-cancer early detection test can detect over 50 types of cancers based on cfDNA. Galleri has a list price of USD $949. A recent GRAIL clinical study by Klein et al. states that the specificity for cancer detection was 99.5%, and sensitivity stood at 51.5% [20]. Noteworthy, the sensitivity increased with stage, with only 16.8% at stage I, 40.4% at stage II, 77.0% at stage III, and 90.1% at stage IV [20]. Looking into more details, in 12 pre-specified cancer, the sensitivity of the test shows a wide variation depending on the cancer type, from 11.2% (prostate), 30.5% (breast), 74.8% (lung), 83.7% (pancreas), and 93.5% (liver/bile-duct) [20]. As semi-expected for any new technology development, the clinical research data from GRAIL have sparked debate as to whether cfDNA/ctDNA technologies could be practical in the clinic [21]. Pons-Belda et al. emphasized that the most important parameter from the patient and physician’s points of view for a cancer screening test is the positive predictive value of the test (PPV) [21,22]. PPV is the risk value of having cancer if the test is positive. The PPV depends on three parameters: test sensitivity, specificity, and disease prevalence within the screened population [21]. Due to technology limitations, Pons-Belda et al. argued that screening for cancer using cfDNA/ctDNA would suffer from low sensitivity and low PPV. GRAIL has since signaled its intention to start an extensive clinical research study to validate its technology.

### 3.3. Circulating Extracellular Vesicles (EVs) for Early-Stage Multi-Cancer Detection

One promising approach to early-detect cancers is based on biomarkers found in circulating extracellular vesicles (EVs) [23]. In a pilot clinical study, Hinestrosa et al. developed an EV-based blood biomarker classifier from extracellular vesicle protein profiles to detect stages I and II of pancreatic, ovarian, and bladder cancer [24]. Notably, the sensitivity detection of stage I cancer was 95.5% in pancreatic, 74.4% in ovarian, and 43.8% in bladder cancer [24].

## 4. Non-Invasive Early Cancer Detection Using Body Fluids

### 4.1. Urine as a Biomarker in Cancer

Early-stage cancer may be detected by computed tomography (CT) and magnetic resonance imaging (MRI), but the use of these techniques is precluded by their high cost and is burdensome. MCED tests may offer a new avenue for the easiest and cheaper cancer screening tests. Still, emerging technologies such as those based on the detection of cfDNA/ctDNA may need more time to mature, especially for early detection at stages 0-I, and become less expensive. A GRAIL Galleri test at USD $949 (about 129,300 JPY) may not resonate favorably for most interested in a regular and affordable multi-cancer screening. Due to low specificity at early stages, GRAIL Galleri may not position itself favorably within MCEDs, especially for identifying the cancer type early on, where it matters the most. Therefore, a more straightforward, non-invasive examination is needed to detect early-stage cancer with less economic burden and physical stress. Non-invasive cancer detection using body fluids such as urine to detect volatile biomarkers is an exciting alternative and is less invasive than blood samples [25]. Interestingly, urine may also become a valid alternative to using blood for detecting and analyzing ctDNA [26]. Rapid developments have been observed in recent years in exploiting molecular biomarkers in urine to develop cancer screening and diagnostic tests, from using analytical techniques to exploiting living organisms such as nematodes or even ants [25,27,28,29]. 

Aside from urine, using tear samples may be interesting for the early detection of breast cancers. A recent study by Daily et al. identified several protein markers in tears for early-stage breast cancer [30]. However, collecting tear samples may be more difficult than collecting urine samples.

### 4.2. The Smell of Cancer and Volatile Urinary Compounds

Cellular transformation and tumor development are associated with dramatic cellular and extracellular modifications. Metabolic signatures of the disease are found in urine. Notably, patterns of volatiles organic compounds (VOCs) are produced by cancer cells and found in urine. A growing number of studies are linking different cancers to either an increase or decrease in levels of VOCs [18,21,26,31,32,33]. It is associated with the cancer smell, the end-product of metabolic changes producing patterns of volatile organic compounds (VOCs) observed in cancers [34,35,36]. Dogs trained with olfactory associative learning are known to recognize specific smells in urine for detecting human lung and breast cancer [37,38,39,40]. Specifically for breast cancer, the study by Woollam et al. identified six VOCs in mice via gas chromatography-mass spectrometry QTOF that could potentially be the basis of screening test with suitable separation of cancer/non-cancer with ROC AUC of 0.96 [41]. Colorectal cancer is the most frequent cancer in western countries, and significant efforts are made to detect it early. Screening and diagnosis are made using a fecal occult blood test, but it requires the collection of a stool sample, which is troublesome and uncomfortable. A test based on a urine sample may be a suitable alternative, as metabolome signatures and specific VOCs have been identified in the urine for colorectal cancer [42]. Many recent studies have highlighted the application of urinary volatile organic compounds found in urine for cancer type discrimination, such as prostate, renal, breast, and lung, among others [39,43,44,45]. 

### 4.3. Electronic Nose (e-Nose)

The first electronic nose with chemical sensors able to detect patterns of chemicals was first announced in 1982 [46]. For the last two decades, in parallel to discoveries of specific VOCs associated with cancer found in urine and exhaled breath, technologies on electronic noses to detect these VOCs have seen rapid development [47,48]. The volatilome is complex, and the e-nose must be sensitive, specific, and selective [49,50,51]. While the link between the volatilome in body fluid, exhaled breath, and diseases continues to be explored and further validated, efforts are underway to develop e-nose technologies for the clinic [52,53,54]. 

## 5. *Caenorhabditis elegans* as a Powerful Biosensor for Cancer Screening

### 5.1. Chemotactic Response of C. elegans with Urine Samples

Many animals rely on olfaction for their survival. The nematode *C. elegans* has a refined sense of smell to finely detect changes in odorant concentrations in its natural habitat to exhibit chemotaxis toward bacteria and its food [55,56]. *C. elegans* has ~1200 olfactory receptor-like genes (~800 for dogs) and approaches a preferred odor while moving away from a disliked odor [57]. *C. elegans* has about 1.5 times as many different types of olfactory receptors as a dog and can detect and respond to extremely low chemical concentrations [58]. Strikingly, Hirotsu et al. demonstrated that *C. elegans* is attracted to urine from cancer patients but tends to avoid urine from healthy persons [59]. *C. elegans* is known to respond to volatile odors through AWA, AWB, and AWC sensory neurons and shows a distinct chemotactic response toward the urine of an individual with cancer compared to healthy ones (Figure 1) [59]. AWA and AWC neurons cause attractive behavior, while the AWB neuron mediates repellent behavior [57,60]. 

### 5.2. N-NOSE

Chemotaxis response of *C. elegans* is concentration dependent, as observed for urine samples [59]. Taken all together, these findings are the foundation of the N-NOSE multi-cancer primary cancer screening test, which is highly sensitive at early stages [13,28,29,59,61,62,63,64]. N-NOSE (Nematode Nose) is an innovative cancer screening test based on the chemotaxis index of *C. elegans* (performed by automated robots) performed at multiple concentrations of urine samples, frozen within two hours of collection [13]. N-NOSE by Hirotsu Bio Science has been sold in Japan since 2020 and costs JPY 14,800 (USD 109). N-NOSE detects the presence of cancer for 15-types: stomach, colorectal, lung, breast, pancreatic, liver, prostate, uterine, esophageal, gall bladder, bile duct, kidney, bladder, ovarian, and oral/pharyngeal (web: hbio.jp, accessed on 2 August 2022). 

### 5.3. Cancer Type Identification with N-NOSE

*C. elegans* is a well-studied model organism, and its neuroscience of olfaction, especially on GPCRs in amphid neurons, has been the focus of many studies. Leveraging on this knowledge, ongoing R&D at Hirotsu Bio Science is producing nematodes showing a specific phenotype for a cancer type using urine [65]. N-NOSE is a primary multi-cancer screening test, but it cannot inform on the type of cancer or staging. An evolution of N-NOSE tests is being developed to pinpoint the cancer type using urine samples, starting with pancreatic cancer [65]. Pancreatic cancer is the 12th most common cancer worldwide but remains one the deadliest cancer, with a 5-year relative survival rate of only 11%, all stages combined. The lack of obvious clinical signs at the early stages renders pancreatic cancer a silent killer [66]. Therefore, a routine pancreatic cancer screening test, sensitive at an early stage, is needed. Differences exist among cancers in the human volatilome in the urine [45,67,68,69]. We hypothesized that knocking out in AWC olfactory neurons of *C. elegans*, a specific GPCR may lead to a chemotactic response specific to a single cancer type using a urine sample. It leads us to the important discovery of cr-4 GPCR in AWC neurons, which is key for a specific chemotaxis response to the urine of patients with pancreatic cancer [65]. Taken together, our findings are the basis of an evolution of the N-NOSE test that is specific for pancreatic cancer at early stage 0-I. Commercial application of N-NOSE cancer type identification for pancreatic cancer starts in late 2022.

### 5.4. N-NOSE Clinical Research

Clinical research was performed at the Engaru Kosei General Hospital (Hokkaido, Japan) between June 2017 and May 2019 to evaluate the accuracy of N-NOSE combined method with 10^−1^ and 10^−2^ dilutions of urine samples [13]. The cohort size was 175 participants, 143 healthy donors, and 32 cancer patients. The sensitivity of N-NOSE was found to be 87.5%, and the specificity was 90.2% [13]. N-NOSE was found to correlate only with the presence/absence of cancer [13]. No significant relationship was found between sex, non-cancer comorbidities, hepatic function, renal function, urine glucose, urine protein, urine ketone bodies, or urine occult blood with N-NOSE results [13]. Importantly was the notable high sensitivity (100%) at stages 0-I [13]. This is in stark contrast to blood tumor markers CEA, CA15-3, and CA19-9 or liquid biopsy approach based on cfDNA/ctDNA such as Galleri by GRAIL (16.8% at stage I) (Figure 2) [21,70].

Cancer surveillance and monitoring response are important for effective cancer therapy. A clinical study was performed at the Nanpuh Hospital (Kagoshima, Japan) between June 2016 and February 2018 to investigate the usefulness of N-NOSE as a postoperative tool for monitoring the presence of cancer for both colorectal and gastric cancers [64]. In this study, 78 cancer patients were enrolled, 46 for colorectal cancer and 32 for gastric cancer. ROC analysis showed that N-NOSE was more accurate than CEA or CA19-9 in correlating with the removal of cancerous tumors [64]. Notably, N-NOSE could discriminate the preoperative and postoperative status even in stages 0 and I. These results suggested that N-NOSE can detect cancer removal regardless of the tumor size [64].

Pancreatic cancer is not the most common cancer, but it remains one the deadliest. Therefore, an affordable, non-invasive, and highly sensitive at early stages pancreatic cancer screening test is sorely needed. Interestingly, Ueda et al., along with the research group of Pr. Ishii at Osaka University (Osaka, Japan) demonstrated that *C. elegans* could recognize the odor of pancreatic cancer in the urine of Kras^G12D^ model mice, suggesting *C. elegans* could also possibly recognize it in the human urine [71]. Next, a clinical study at Osaka University on patients with pancreatic ductal adenocarcinoma (PDAC) between October 2015 and June 2019, with very early-stage pancreatic cancer (stage 0 or IA), showed that *C. elegans* could indeed detect the scent of cancer for the diagnosis of early-stage pancreatic cancer [61]. In this study, 83 PDAC patients were enrolled in an open-label study with pathological stages (0/IA/IB/IIA/IIB/III/IV) to monitor the evolution of the *C. elegans* scent test before and after surgery. Next, 11 PDAC patients at an early stage and 17 healthy volunteers were enrolled in a blind study with the same scent test performed before and after surgery. Preoperative urine samples had a significantly higher chemotaxis index than postoperative samples in patients with pancreatic cancer, with an AUC = 0.845 (10^−1^ dilution of urine) and AUC = 0.820 (10^−2^ dilution), all stages considered, highlighting the suitable performance of the method [61]. By comparison, for early PDAC cases (stage 0 or IA), the protein tumor markers CEA and CA19-9 showed very poor sensitivity at 0% and 27% (blinded trials in stages 0-IA), respectively [61]. In this clinical study, N-NOSE significantly differed between cancer and healthy at 10^−2^ dilution (*p* = 0.034) [61].

A further clinical study to evaluate N-NOSE on pancreatic cancer was performed at Saitama medical university (Saitama, Japan) between July 2017 and February 2019. In this study, 441 persons were enrolled that included 104 pancreatic cancer patients and 95 in the control group without cancer. The sensitivity and specificity of N-NOSE for pancreatic cancer were 84.6% and 60%, respectively, when all stages were combined [62]. At PDAC stage I, N-NOSE had an 87.5% sensitivity, while CEA and CA19-9 markers were at 0% and 28.57% sensitivity, respectively [62]. At stage II, N-NOSE had a 93.3% sensitivity, while CEA and CA19-9 were at 14.29% and 78.57% sensitivity, respectively [62]. Confirming findings from the clinical study at Osaka University, N-NOSE detected stages 0 to I pancreatic cancer and had a higher correlation with early-stage pancreatic cancer than the advanced stages [62]. Similarly, as observed in the clinical study at the Engaru Kosei General Hospital, only N-NOSE correlated with the presence/absence of pancreatic cancer, and bilirubinuria levels (obstructive jaundice may have been caused by pancreatic cancer) had no impact on the positivity of the resulting [62]. 

Several clinical research is ongoing at various medical institutions in Japan but also abroad to continue evaluating the suitable performance of N-NOSE.

### 5.5. Replication of N-NOSE Behavioral Assay by Other Research Groups

The advent of N-NOSE as a *C. elegans* behavioral assay with urine samples has prompted interesting studies from other research entities that replicate and validate N-NOSE core findings. Recent clinical research on prostate cancer from the Oregon Health & Science University and Providence St. Joseph Health (Portland, United States) was able to confirm the findings of Hirotsu et al. about *C. elegans* being attracted, in a dilution manner (10^−1^ and 10^−2^), to the urine of patients with prostate cancer [29,59]. A clear chemotactic behavioral difference was observed between the healthy group and the prostate cancer group, as observed for N-NOSE [29].

A recent clinical study by the Italian Institute of Technology and the Mother G. Vannini Hospital (Rome, Italy) on breast cancer was also able to replicate N-NOSE findings, with *C. elegans* showing clear chemotaxis index differences between healthy and breast cancer groups, using diluted urine samples at 10^−1^ and 10^−2^ dilution [28].

## 6. Conclusions: N-NOSE as a Multi-Cancer Primary Cancer Screening Test

The advent of MCED tests and large-scale cancer screening at tighter intervals may induce some stress on the healthcare system. Undoubtedly, the healthcare system will have to adapt to cope, but the benefit of MCED tests possibly outweighs the drawbacks. N-NOSE has a marked advantage against protein tumor markers, cfDNA/ctDNA-based technologies, and the classical one-test one-cancer screening approach. The cost of N-NOSE (JPY 14,800 or ~USD $109) for 15-types of cancer with high sensitivity at early stages is significantly cheaper than Galleri from GRAIL (USD $945) for 50-types of cancer and which has a lower sensitivity at early stages. N-NOSE has the potential of being a multi-cancer primary cancer screening test for early detection but also be important for precision medicine. N-NOSE is a *C. elegans* behavioral test and is cheaper technology to operate than cfDNA/ctDNA-based technologies and easier to scale up for large-scale cancer screening. At the time of this writing, over 200,000 persons were screened by N-NOSE in Japan.

## Figures and Tables

**Figure 1 biomedicines-10-02371-f001:**
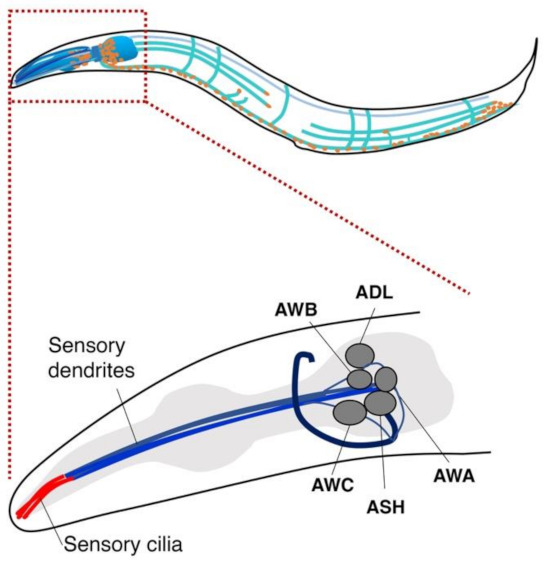
Five pairs of olfactory neurons in *C. elegans*. A variety of odorants are received by olfactory receptors in five pairs of neurons, AWA, AWB, AWC, ASH, and ADL, in *C. elegans*, which render worms to cause specific chemotaxis behaviors. AWA and AWC neurons are known to be involved in attractive behavior, while the AWB neuron causes repellent behavior.

**Figure 2 biomedicines-10-02371-f002:**
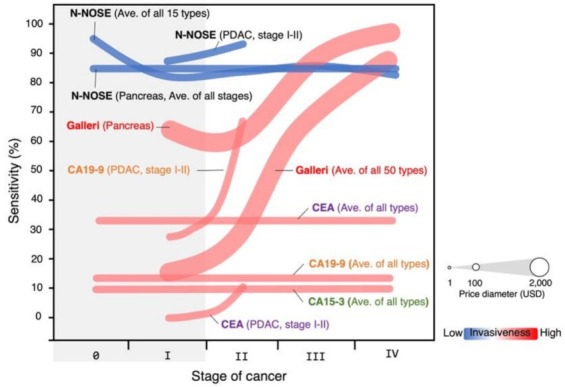
Comparison in sensitivity between N-NOSE and the other existing cancer screening kits. The diameter indicates the price (USD) of the kit. The color heat map indicates the invasiveness of the kit. N-NOSE is a less expensive and non-invasive screening kit with remarkable sensitivity for early-stage (0-I) cancers.

## Data Availability

Not applicable.

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
