# Peer review of "C. elegans as a Powerful Tool for Cancer Screening"

_biomedicines, 2022, doi:10.3390/biomedicines10102371_

Round 1
Reviewer 1 Report
In this review paper, the authors show the progress that has been made in multi-cancer early detection (MCED) as a potentially effective routine cancer screening test. The results with a test based on the chemotaxis of C. elegans using a urine sample (N-NOSE) are impressive with 87.5% sensitivity and 90.2% specificity for detection of early cancer. This N-NOSE test is already a commercially available primary cancer screening test in Japan.
The need for a minimally-invasive and relatively inexpensive multi cancer diagnostic test is obvious: simultaneous detection of multiple cancers, possibly at early stages, probably resulting in a drastic reduction in the number of patients with late stage disease and cancer related deaths. It may improve public health by limiting the need for cumbersome and invasive diagnostic tests and may reduce healthcare costs.
The N-NOSE may become a paradigm shift in cancer screening, since currently most people do not seek a cancer screening test without any sign of physical disorder. Due to the lack of clinical signs in early stages, cancers are usually detected in a relatively late phase.
The main advantage of the N-NOSE is its high sensitivity for very early cancers (higher than for advanced cancers), the authors even acclaim 100% sensitivity for gr 0-1 early cancers. Other existing MECD methods have higher sensitivity for more advanced cancers.
The results with N-NOSE as an effective primary cancer screening test are impressive, but next steps that have to be taken in case of a positive test are unclear. As with other MECD-tests, N-NOSE does not point to one specific disease, but to 15 different cancer types. For accurate treatment, confirmation of the diagnosis and localization of the disease is essential. And for certain types of early cancer (e.g. pancreatic cancer) this confirmation is hard to obtain, and the positive predictive value of the test may be difficult to assess. Moreover, a positive test will imply a significant burden for the health system to confirm or exclude the presence of specific types of cancer, and it will have large emotional and psychological consequences, since total sensitivity and specificity of the test are not 100%.
Before widespread screening in the population, this next step should be clearly defined and more cancer-type specific non-invasive tests should be developed to narrow the non-specific cancer suspicion. The next step will involve "one test, one cancer" principle, a more tumor-specific minimally-invasive and inexpensive diagnostic test. Possibly, the authors may "train" nematodes to like the scent of more tumor-specific volatile organic compounds.
In hereditary cancer syndromes, however, with individuals at increased risk of developing specific cancers, the N-NOSE test may become an excellent tool for non-invasive longitudinal surveillance.
I suggest that the authors might address these issues in their excellent overview.
Author Response
Response: Thank you very much for your positive and constructive comments. We have revised, edited for clarity, expanded, and improved the manuscript according to the reviewers' comments.
Practically, if an N-NOSE result shows an elevated risk of the presence of cancer, we have established a procedure to guide customers to the next step, taking into consideration emotional and psychological consequences. We have also established a system where customers can discuss directly with medical doctors about positive N-NOSE results and what to do next. In this review, we do not discuss customer management and operational guidelines, as we only focus on the scientific aspect of using C. elegans for cancer screening.
The advent of MCED tests and large-scale cancer screening at tighter intervals may induce some stress on the healthcare system. Undoubtedly, the healthcare system will have to adapt to cope, but we reckon that the benefit of MCED tests outweighs the drawbacks.
We have clarified the concept of N-NOSE as a primary cancer screening test and added a new section about the next evolution of N-NOSE that is cancer-specific (section 5.3 lines 236-254). N-NOSE is a primary screening test. In case of a positive result, the customer can choose to go to the next step, to pinpoint the cancer type, with an evolution of N-NOSE that is cancer-type specific. We have genetically modified C. elegans (on specific GPCR receptors) that trigger specific chemotaxis with the urine of specific cancer (lines 236-254). The first next-generation N-NOSE test is pancreatic-cancer specific (at early stage 0-I), and we are planning to start business usage later this year. In the coming months, we plan to release several next-generation N-NOSE tests to pinpoint other types of cancer specifically.
Reviewer 2 Report
The manuscript by Hirotsu et al. reviews focus mostly on their published work in the development of N-NOSE technology. From the reviewer standpoint, this manuscript lacks critical revision on the available technologies for pan-tumour detection, particularly the ones opposing the detection of tumour-associated metabolite compounds (either by N-NOSE or other type of e-NOSE tech). No critical discussion regarding innovative technologies as circulating EVs (e.g. PMID 32795414), cfDNA (e.g. reviewed in PMID 33473219) is provided. Similarly, a critical discussion on how N-NOSE compares with other e-NOSES for the recognition of tumour-associated volatile organic compounds in samples for the detection of tumours of unknown origin is also lacking. Thus, the reviewer does not recommend the publication of this manuscript in Biomedicines journal.
Author Response
Response: Thank you for your comments. We have revised thoroughly the manuscript, edited it for clarity, and augmented it considering all reviewers’ comments. This review focuses on C. elegans olfaction and the volatilome in urine. We also mention and discuss other technologies for MCED-tests already on the market. We have included critical discussion regarding other innovative technologies. Section 3.2 (lines 142-148) outlines cfDNA technology by GRAIL and we have included a new paragraph (section 3.3, lines 142-148) on circulating EVs. We have also included a critical discussion about electronic noses on section 4.3 (lines 195-203).
Reviewer 3 Report
I read with great interest the review paper by Di Luccio et al, which highlighted C. elegans as a promising tool for cancer screening due to its high sensitivity and specificity demonstrated in previous studies. Importantly, this nematode has become a popular model for biological and basic medical research. It has also been regarded as an ideal model system for high-throughput drug screening and a model system for investigating fundamental questions in various areas of biology, including tumorigenesis. The present paper refers to the C. elegans (N-NOSE) application as a multi-cancer screening test. Overall, this is a very interesting and promising topic in cancer and the paper is written in a very clear and sequential manner.
I have just a few concerns that are listed below:
- I suggest improving chapter 1 to make it clearer which cancer screening tests are routinely recommended as standard of care for all patients.
- Diagnosis by canine olfaction, using dogs trained to detect cancer by smell, has long been shown to be both specific and sensitive. Similarly, C. elegans uses olfaction for cancer detection based on breath odor. So, does C.elegans have a more developed olfactory system than dogs? Do the authors have any explanation for the high accuracy of N-NOSE in the early stages of cancer?
- Many recent studies have highlighted the application of urinary volatile organic compounds (VOCs) in different types of cancer urine discrimination, such as prostate (DOI: 10.3390/cancers12082017), renal (DOI: 10.1111/jcmm.13132), breast (DOI: 10.1038/s41598-022-17795-8), and lung (DOI: 10.1186/s12885-021-08651-5), among others. This should be mentioned in the manuscript to support the N-Nose model's potential for cancer screening.
- The authors mention using N-NOSE for multi-cancer screening tests, but is this limited to 15 cancer types? Furthermore, does this imply that the cancer is present or not but no one knows what type it is?
- Current research limitations in this field and future directions should be presented and discussed. For example, the majority (if not all) clinical studies have been conducted in Japan and have not been validated in different patient cohorts. Do the authors believe N-Nose can be used as scalable diagnostic sensors?
Minor:
-Please indicate the number of people who participated in the aforementioned clinical studies.
-Subtitle 5.3, First paragraph, and Figure 2- Please clarify the cancer type so that different cancer screening tests can be compared.
Author Response
“I read with great interest the review paper by Di Luccio et al, which highlighted C. elegans as a promising tool for cancer screening due to its high sensitivity and specificity demonstrated in previous studies. Importantly, this nematode has become a popular model for biological and basic medical research. It has also been regarded as an ideal model system for high-throughput drug screening and a model system for investigating fundamental questions in various areas of biology, including tumorigenesis. The present paper refers to the C. elegans (N-NOSE) application as a multi-cancer screening test. Overall, this is a very interesting and promising topic in cancer and the paper is written in a very clear and sequential manner.
I have just a few concerns that are listed below:”
- I suggest improving chapter 1 to make it clearer which cancer screening tests are routinely recommended as standard of care for all patients.
Response: Thank you for your positive and constructive comments. We have worked out the whole manuscript for enhanced clarity. We have clarified and augmented chapter I and added screening tests routinely recommended (lines 20-52).
- Diagnosis by canine olfaction, using dogs trained to detect cancer by smell, has long been shown to be both specific and sensitive. Similarly, C. elegans uses olfaction for cancer detection based on breath odor. So, does C.elegans have a more developed olfactory system than dogs? Do the authors have any explanation for the high accuracy of N-NOSE in the early stages of cancer?
Response: Thank you. We have clarified and augmented paragraph 5.1 (lines 204-217) about C. elegans olfaction to precise the olfaction of C. elegans compared to dogs. In brief, C. elegans is blind and relies on its extremely attuned sense of smell to navigate and feed itself. C. elegans has ~1,200 olfactory receptor-like genes (~800 for dogs). That is about 1.5 times as many different types of olfactory receptors as a dog and can detect and respond to extremely low chemical concentrations lower than trained dogs can detect. Taken all together, it is the most probable explanation for the high accuracy of N-NOSE at early cancer stages.
- Many recent studies have highlighted the application of urinary volatile organic compounds (VOCs) in different types of cancer urine discrimination, such as prostate (DOI: 10.3390/cancers12082017), renal (DOI: 10.1111/jcmm.13132), breast (DOI: 10.1038/s41598-022-17795-8), and lung (DOI: 10.1186/s12885-021-08651-5), among others. This should be mentioned in the manuscript to support the N-Nose model's potential for cancer screening.
Response: Thank you very much for these good suggestions. We have added these to the manuscript (lines 191-193).
- The authors mention using N-NOSE for multi-cancer screening tests, but is this limited to 15 cancer types? Furthermore, does this imply that the cancer is present or not but no one knows what type it is?
Response: Thank you. Current N-NOSE is a MCED test, or primary cancer screening test that can detect the presence of cancer with much more sensitivity at early stages than blood tumor markers, for instance. The advantage of N-NOSE (and MCED tests) is that a single simple test, non-invasive, can inform on the early presence of cancer. N-NOSE has been proven for 15-types of cancers. In case of an N-NOSE result showing a likelihood of the presence of cancer, the next step is to identify the type of cancer by classical methods at the hospital. However, we are developing the next generation of N-NOSE, that will be able to identify the cancer type specifically. In that case, the first step in cancer screening is N-NOSE for detecting the presence of cancer, followed by N-NOSE cancer-type identification to pinpoint the type of cancer. We have clarified the manuscript and added a new section about the next-generation N-NOSE tests (section 5.3 lines 236-254).
- Current research limitations in this field and future directions should be presented and discussed. For example, the majority (if not all) clinical studies have been conducted in Japan and have not been validated in different patient cohorts. Do the authors believe N-Nose can be used as scalable diagnostic sensors?
Response: Thank you. Our first clinical research on N-NOSE done in Australia indicated that ethnicity doesn’t correlate with N-NOSE results. In other words, N-NOSE works for everyone. Currently, more than 20 clinical studies are being undertaken at different medical institutions in Japan, but also abroad, with a large clinical study being performed at the Queensland University of Technology in Australia. In the coming months and years, regularly, we will be publishing clinical research results done in Japan and abroad. We have mentioned it in this revised manuscript on lines 318-319.
N-NOSE is fully scalable and is a commercial hit in Japan. As we speak, more than 200,000 customers have taken N-NOSE tests. We operate N-NOSE in three facilities in Japan. N-NOSE is fully automated with custom-made robotic lines that automatically process samples and perform chemotaxis assays. Since the start of N-NOSE business in 2020, we have regularly added capacity (more robotic lines) to cope with the steady increase of demand.
- Minor:Please indicate the number of people who participated in the aforementioned clinical studies.
Response: Thank you. Yes, we have added this information.
- Minor: Subtitle 5.3, First paragraph, and Figure 2- Please clarify the cancer type so that different cancer screening tests can be compared.
Response: Thank you. Yes, we have modified figure 2 to address this valid concern.
Round 2
Reviewer 3 Report
The authors have addressed all of my concerns, and I believe the article can be accepted in its current form.